# Shotgun Metagenomics Reveals Minor Micro“*bee*”omes Diversity Defining Differences between Larvae and Pupae Brood Combs

**DOI:** 10.3390/ijms25020741

**Published:** 2024-01-06

**Authors:** Daniil Smutin, Amir Taldaev, Egor Lebedev, Leonid Adonin

**Affiliations:** 1Institute of Environmental and Agricultural Biology (X-BIO), Tyumen State University, Tyumen 625003, Russia; 2Faculty of Information Technology and Programming, ITMO University, St. Petersburg 197101, Russia; 3Institute of Biomedical Chemistry, Moscow 119121, Russia; 4Research Center for Molecular Mechanisms of Aging and Age-Related Diseases, Moscow Institute of Physics and Technology, Dolgoprudny 141700, Russia

**Keywords:** honey bee *Apis mellifera*, metagenome, shotgun sequencing, hive, comb, bee larvae, bee pupa, bacterial diversity, symbiosis

## Abstract

Bees represent not only a valuable asset in agriculture, but also serve as a model organism within contemporary microbiology. The metagenomic composition of the bee superorganism has been substantially characterized. Nevertheless, traditional cultural methods served as the approach to studying brood combs in the past. Indeed, the comb microbiome may contribute to determining larval caste differentiation and hive immunity. To further this understanding, we conducted a shotgun sequencing analysis of the brood comb microbiome. While we found certain similarities regarding species diversity, it exhibits significant differentiation from all previously described hive metagenomes. Many microbiome members maintain a relatively constant ratio, yet taxa with the highest abundance level tend to be ephemeral. More than 90% of classified metagenomes were Gammaproteobacteria, Bacilli and Actinobacteria genetic signatures. Jaccard dissimilarity between samples based on bacteria genus classifications hesitate from 0.63 to 0.77, which for shotgun sequencing indicates a high consistency in bacterial composition. Concurrently, we identified antagonistic relationships between certain bacterial clusters. The presence of genes related to antibiotic synthesis and antibiotic resistance suggests potential mechanisms underlying the stability of comb microbiomes. Differences between pupal and larval combs emerge in the total metagenome, while taxa with the highest abundance remained consistent. All this suggests that a key role in the functioning of the comb microbiome is played by minor biodiversity, the function of which remains to be established experimentally.

## 1. Introduction

### 1.1. Bee Hive as Microbiology Model Object

Microbiomes, as well as any other communities, are closely related to each other within a biocenosis [1,2]. Functioning together, they form an environment. Some microorganisms may occupy more than one ecological microniche and can disrupt the functioning of microbiomes by inherently becoming introduced species [3,4,5]. Therefore, the stability of communities is determined by their complexity and complex genomic signatures—metagenome [1,6,7,8]. Functional differences and host genomics influence between microbiomes with similar conditions are often determined by minor representatives of communities [9,10,11]. Even these small variations in microbiota mediate big proteomics, metabolomics and full microbiome composition changes [11,12].

Bee colonies form an environment for a distinct microbiota [13,14]. Hive members, their microflora and microorganisms living on different hive substrates influence each other and function together as a superorganism [13,15,16]. The microbiota influences the metabolism [13,14,15,17], development [18,19,20], immunity [21,22,23,24] and even behavior [25,26,27] and evolution and speciation [28,29] of bees. The gut microflora of bees and its role are well studied [13,15,30,31]. In adults, it includes only nine major groups of bacteria that populate the gastrointestinal tract during the first days after hatching from the pupae [13]. The pupal intestine contains almost no bacteria, which makes bees a good object for setting up model experiments to study the microbiome in adults [13,30]. The other components of the hive microbiome, such as the comb microbiome, have received far less research attention, with traditional culture methods typically being the primary approach [15,16,31,32,33]. However, even this small number of articles indicates the huge role of the hive microbiome in the life of its inhabitants [15,23,34,35]. It is the bee hologenome, which includes the genomes of all hive organisms, that should determine its characteristics [31,32].

### 1.2. Bee Hologenome Parts

Hive environments contain three functional parts of microbiota. Some bacteria can be transposed between hive members from different hive substrates [36]. Worker bees transfer bacteria and fungi from plants and air into the hive [35,37]. The main part, normal microflora, protects the hive environment from undesirable and pathogenic invaders [23,24,33]. Also, it is involved in hive metabolism systems [16,37,38]. *Apilactobacillus kunkeei*, *Bombella apis* and other species play a role in honey and pollen processing [16,38].

Honey contains a large number of microbes [14,35,39,40,41]. The composition of species within the hive environment is influenced by two primary factors: firstly, it is contingent upon the floral sources from which the bees forage for nectar [35,40]; and secondly, it undergoes a progressive alteration throughout the process of honey maturation [39]. Bee bread, bee pollen and propolis also contain bacteria involved in their formation [33,34,38,42,43,44]. These sugar-rich substrates, however, maintain a fairly constant microflora [16,35,37,43,44]. This is controlled by a multitude of antimicrobials [34,45,46,47], which also influence other microbiota communities of hive [44,48,49,50] and landscape [16,51], and bees themselves [19,34,37,52,53]. Like in most hive niches, the majority of studies report domination of bacteria, especially *Bacillus*, *Lactobacillus*, *Apilactobacillus* and *Bifidobacterium* [16,35,37]. Lipidome and shotgun metagenomics studies predict higher amounts of fungi than expected in previous articles [41,43,53,54].

Comb microbiota community might be similar to honey, but less rich [15,54]. Previous articles had reported low biomass and diversity of microorganisms of brood combs [15,54,55,56,57]. Cultural methods reveal that combs hives contained only 10–106 CFU/g [55]. Investigated brood combs recovered less diversity and biomass than common hive floors, especially in spring and winter. At least eight *Bacillus* species, *Lactobacillus*, *Apilactobacillus*, *Brevibacillus*, *Corynebacterium*, *Leifsonia*, one Enterobacteriaceae group and several Actinobacteria were identified in combs [55,56,58]. Amount of microorganisms on hive floors are higher [28], while they contain notably lower concentrations of nutrients [16]. So, there might be a mechanism of comb microbiota control.

## 2. Results

### 2.1. Taxonomy Annotation

Approximately two million Illumina shotgun reads per sample were sequenced across a dataset comprising 17 samples—equivalent to a cumulative data corpus of about 35 million reads. More than 95% of all reads were identified as *Apis mellifera*, and more than 80% of remaining reads were not classified.

Only 2.1% of all reads were classified on the HoloBee database [59] (Appendix A). Despite results on the RefSeq database, it well identifies DNA from different metazoans, but their numbers fluctuate considerably (Appendix A).

Bacterial abundance and diversity (more than 260k reads, 9000 species, 1497 genus) significantly exceed total viral (6.7k reads, 79 species, 59 genus) and fungi (8k reads, 96 species, 258 genus) (Figure 1). It is likely that some of the taxa were misclassified (Appendix A). Among the dominant species, this applies primarily to *Buchnera* and *Plantactinospora*. These classifications most likely refer to other representatives of Enterobacteriaceae and Micromonosporaceae, respectively. Therefore, henceforth we regard them as putative taxa in relation to these genera exhibiting clustered dissimilarities among metagenomes.

Most numerous fungal groups were Ascomycota, Basidiomycota and Mucoromycota (Appendix A). Of the 258 fungal genus, a minimum of 223 are classified by multiple reads from more than two samples. No species dominates that microbiome part.

Some bee pathogens also were detected. All samples were infected by *Varroa destructor* and some also by *Apis mellifera filamentosus virus* (*AMFV*) (Figure 2).

All analyses show a fairly high taxonomic diversity. Major groups by amount are Actinobacteria (*Plantactinospora*, *Streptomyces* and unclassified taxa), Gammaproteobacteria (mainly *Gilliamella* and *Pseudomonas*), Lactobacillales (*Streptococcus*, *Apilactobacillus* and *Lactobacillus*) and Bacillales.

Of the widely represented microorganisms, only representatives of the genera *Bacillus*, *Apilactobacillus* and *Lactobacillus* have been previously isolated from brood combs. Representatives of *Corynebacterium* and *Brevibacillus*, whose biodiversity in the hive has been described by culture and biochemical methods, were observed in small numbers.

The greatest number fluctuations between samples are characteristic of *Gilliamella*, *Apilactobacillus*, *Serratia* and *Parasacharibacter* (Appendix A). Other bacteria are classified in the metagenome in almost constant amounts.

Analysis performed on HoloBee database discovers high levels of *Aethina* in the sample majority (Appendix A). Classification of *Gilliamella* and *Apilactobacillus* remain similar with analysis on the RefSeq database, whereas other Bacteria taxa are classified badly.

The level of *Varroa destructor* DNA in the metagenome was also constant. Eight samples contain *AMFV* reads (Figure 2). This plot shows no correlation between virus infection and metagenome member amount.

The composition of bacterial and fungal microbiomes in the combs is fairly constant (Figure 2 and Appendix A). A large number of *Gilliamella* in samples 4, 7 and 17 and *Apilactobacillus* in 2, 5 and 6 are isolated. At the same time, the ratios of other groups remain close to constant.

In further analysis, we used bacteria, virus, and HoloBee Kraken 2 and Kaiju fungal taxonomies.

### 2.2. Diversity Analysis

The alpha-diversity analysis is presented on Appendix A. The dominance analyses show that 2–4 samples fall out of the total sample depending on the analysis. All of these samples show large amounts of *Gilliamella* and Protobacteria in general, which may have been introduced from the gut of adults. Evenness analyses isolate these same samples, or describe all samples homogeneous. In contrast, the remaining samples do not exhibit significant differences from each other in terms of these metrics.

Different richness measures and Kempton and Taylor’s Q coefficient differ in the other two samples, from different hives and at different stages. The sizes of these metagenomes do not differ from the others, similar numbers of *Apis mellifera* reeds were isolated in these samples, and their metagenomic compositions are similarly typical.

Jaccard’s dissimilarity does not show any grouping of samples by overall diversity (Appendix A). The individual difference between the bacterial composition of the samples is greater than between any groups. The coefficient values range from 0.63 to 0.77, indicating a high consistency in bacterial composition. The composition of viruses and fungi differs to a greater extent. This is due, on the one hand, to the fact that fewer of these taxonomic units are annotated in the samples. On the other hand, these groups are mostly represented by few species with stochastic occurrence in the samples.

Bray–Curtis dissimilarity separates samples by hive and sampling stage (Appendix A). Moreover, it is most pronounced for the coefficient calculated on virus diversity. It is worth noting that the only detected pathogenic *Apis mellifera virus*, *AMFV*, is more prevalent in the first hive and among larvae (Figure 2). Although individual differences in fungal diversity between samples are greater than group differences, they cluster in a similar manner. Bacterial and total diversity separated samples by hive and stage to a lesser extent. The three samples falling out of the total sample by alpha diversity form a separate cluster.

### 2.3. Composition Analysis

To identify the relationships between microbiome members, cluster (Appendix A), correlation (Appendix A) and tSNE (Appendix A) analyses of the most represented bacterial genera were performed. The results of these analyses are summarized in Figure 3.

Most of the diversity forms a single cluster (Appendix A). Bacillaceae, Lactobacillaceae and Enterobacteriaceae are most widely represented in it. Separating from this cluster are *Streptococcus* and *Plantactinospora*, *Streptomyces*, *Pseudomonas* and *Buchnera*, and separately, *Gilliamella* and *Apilactobacillus*.

A correlation analysis showed more distinct clusters. No significant correlations were found for *Apilactobacillus*, *AMFV* and *Parasacharibacter* (Appendix A). For some other species, correlation patterns seem to be random.

The most abundant taxa form five distinctive groups (Figure 3). Members of the bee gut microbiome *Gilliamella*, *Frishella*, *Snograsella* [13], *Bartonella* [60] and *Parasacharibacter* [61] show similar patterns of correlation with other species and their numbers in the metagenome are highly correlated with each other. This cluster could be divided in two subgroups—bigger “gut” include core adult gut members, and smaller “*Bartonella*” have higher correlations among them and slightly lower between.

The most represented, Actinobacteria, Bacillaceae and Enteobacteriaceae, form separate clusters we named “ABE”. Their composition is significantly negatively correlated with the “gut” cluster, but not with the “*Bartonella*” group. The strongest negative correlations in this cluster are observed for *Bacillus*, *Streptomyces* and *Plantactinospora*.

Another cluster includes potentially pathogenic bacteria—*Pseudomonas* [62,63] and *Corynebacterium* [64]—also found in small numbers in guts. However, it may include different members of *Pseudomonas* that differ in their ecology in the hive [65]. Its abundance correlates weakly with the other clusters.

*Spiroplasma* and *Serratia*, previously found in bees [66,67], with some other bacteria form two clusters (“S+S”). They do not correlate with each other very much, while the correlation patterns with other clusters are similar. Members of this cluster also could be potential pathogens. Bee louse is also related to them and might mediate transmission [68].

The tSNE compositions show some clustering patterns (Appendix A). The most species amount belongs to the “ABE”, gut and “*Bartonella*” groups, which do not separate from each other in the graph. tSNE shows that other heterogeneity can be observed in this group. The main representatives of these clusters are evenly distributed among this “super”-group in the graph. This may be due to their high constancy in all samples.

*Apilactobacillus*, *Lactobacillus* and *AMFV* form clusters with a small number of species. These signatures may be random, the results of species misclassification, or have some relationship with these species. Numerous species are not found among them.

A separate cluster is formed by *Pseudomonas* while “S+S” genomic signatures are scattered among all diversity. Also, tSNE reveals two mostly Acetobacteraceae groups, and three other clusters with small amount and diversity.

### 2.4. Composition Associations

While metagenome compositions provide valuable insights, it is imperative to also consider their functional significance within the community. To figure this out, we constructed a heatmap of samples across the most represented taxa (Figure 4). The clustering of combs with bees at the same stages and predominantly from the same hive is observed. Microorganisms may be grouped into small clusters according to environmental tolerance.

The heatmap does not form any discernible patterns in sample composition. Classified viruses and other taxa with slight abundances are usually detected in only one sample each, which makes them almost unaffected by clustering. Numbers of fungi and bacteria fluctuate between samples, but not significantly. But in general, samples form clusters by brood stage.

We compared taxa abundances between hive combs with different stages (Appendix A). In both hives studied, higher numbers of *Apilactobacillus* are found in cells with larvae. *Gilliamella* is more abundant in pupal cells, but only in one of the hives. Between brood combs with different stages vary the z-scale of *Klebsiella*, *Snoegrasella*, *Buchnera* and *Streptomyces* amount, while most other numerous members’ z-scaled amount differs between samples (Appendix A). Combs metagenome mappings differ both by stage and by the hive from which they were sampled (Figure 5). Dissimilarity between combs with different stages must be bigger than dissimilarity between combs from different hives. Differences are observed for projections on the second to fourth rather than first component. This possibly indicates an important role of minor diversity.

### 2.5. Genome Annotations

We analyzed gene annotations in the metagenome. Among the known genes, *Gilliamella*, *Bacillus* and *Streptococcus* genes are the most widely represented. The best InterPro annotations were obtained on large scaffolds with an ORF length greater than 100 on the PANTHER database (Appendix A). Most of the predicted genes have a Pollen A allergen signature, and a significant fraction of proteins were simply recognized as hypothetical allergens (Appendix A). Antifreeze, ferredoxin and IGF-like proteins were the most widely annotated.

There were also a lot of genes identified related to energy, cell wall, shikimate and b12 metabolism. Surprisingly, a lot of genes were related to tagatose, petrobactin and molybdopterin metabolism. Also, we find several genes related to antibiotic production and resistance: *macB*, *ycaO*, *cvaA*, *bcr* and *bceA*.

## 3. Discussion

Brood combs provide a medium for a diverse microbiome. All major representatives of the gut microbiome of adult and larvae bees [29,69,70,71] and honey [35,40] were observed in the samples. Bacteria abundance is much higher than eukaryotic. The most abundant bacterial species coincide in most microbiomes. Similar situations are observed in other hive metagenomes [14,32,44,72,73,74]. Metagenomes vary due to environmental factors and control systems. The bee gut microbiome contains only a few species that determine almost all of its functionality and stability [13,17,75]. At the same time, honey [14,39,40], propolis [43,48], royal jelly [24,61,76] and brood cells contain a diverse microbiota without sharply distinguished dominants. Unlike the gut, representatives of these microbiomes can be found outside the hive [15,23,31]. Their specialization and transmission between hives remains an open question [15,72,77]. In fact, some strains and species appear to be specialized only to specific hive microniches [78,79,80], so they must be transduced between hives. Environmental species also adapt to specific ecological factors of the hive and form community parts depending on that and pollination landscape [16,35,37].

More than 18% of taxa in shotgun honey research was identified using a custom database, while HoloBee reveals only 8% of species listed [35]. We investigated hives remote from the usual study sites, and among other reasons, the results of taxonomic annotation are different from those known on bees (Appendix A). The brood comb metagenome shows significant numbers of bacilli, as in the honey metagenome [35], and gamma-proteobacteria, as in larvae at some stages [69,70,71]. Unlike other hive metagenomes, combs are dominated by Actinobacteria (Figure 2).

The number of taxa detected is very much higher than predicted by culture methods [55]. This may be due to the fact that nutrient agar was used in this work. *Gilliamella* and some other representatives of the adult gut are not cultured on aerobic media, although they can be found in the hive under aerobic conditions. *Plantactinospora* grows poorly on nutrient agar, as do other bacteria, especially compared to Bacilli. Actinobacteria genus *Actinomadura*, *Nocardiopsis*, *Nonomuraea* [58] and *Leifsonia* [55], previously identified in hive combs, were not found. Other taxa found in previous studies were also found by us. *Streptomyces*, *Bacillus*, *Lactobacillus*, *Corynebacterium* and *Apilactobacillus* species are represented by a large number of closely related reads in the pooled microbiome. We cannot verify the presence of Enterobacteriaceae enteric group 60 from previous research [55]. It was firstly described as *Morganella* [81] and might be related to hypothetical Erwiniaceae and *Buchnera* or other Enterobacteriales members of comb microbiome.

Bacteria form three groups by their abundance correlations (Figure 3). Most likely, an admixture of adult gut core species representatives is a contamination. *Gilliamella*, *Frishella* and *Bartonella* are rare in hive environments [15,37], obligate anaerobic [13] and do not determine the clustering of samples. *Parasacharibacter apium* that cluster with them are found in the larval midgut and food stores [82]. So, this contamination might be mediated by foragers. *Mycoplasma*, *Lysobacter*, *Vibrio*, Helicobacter and some other bacteria were not previously found in hives. But reads of these hypothetical taxa also cluster with core gut species, so all this group (“gut” and “*Bartonella*”) could be a contamination.

Other big group include the majority of saccharolytic Bacillaceae and Enterococcaceae, Actinobacteria and also gut *Burkholderia* [83] (“ABE” group). *Klebsiella*, *Shigella*, *Esherichia* and other Enterobacteriaceae might have main metabolic functions [84]. Small hive beetle amount correlates positively with members of this group. Some “ABE” Enterobacteriaceae species previously were found as associated with *Aethina* microbiota [59]. All taxa that we also found were not unique for that beetle, so it could play the role of transferring part of brood comb microbiota [59,85]. Majority Bacillaceae and any Actinobacteria from “ABE” were not previously found in association with *Aethina*, so understanding of their transmission and function validation is a future work.

“ABE” group composition is negatively correlated with the abundance of the “gut” group. The introduction of gut microbiota from the hive can be random but should be an ongoing process. Bacilli species should increase stability of microbiome [57]. Actinobacteria and some *Bacillus* and *Streptomyces* species can produce antimicrobials against invaders [46,86,87]. So, in addition to contamination, the “gut” group may include organisms that are in an antagonistic relationship with “ABE” species. They may have lost some of their resistance genes to secreted antimicrobials. Their consistency and function in brood combs should be investigated in future.

Gene annotation analysis identified several known genes including five genes associated with antibiotic synthesis or resistance. MinPath predicts multiple genes for basic metabolism and cell wall synthesis which are common in bacterial metagenomes. A lot of *Gilliamella* genes with well-annotated functions also were detected [88].

Surprisingly a lot of annotated genes were related to tagatose metabolism and possibly to D-tagatose pathway. This sugar may be found in nectar and honey [89], and apparently some members are specialized to metabolize it. This niche may be occupied by some *Staphylococcus* or *Bacillus* strains in which a relevant metabolism has been previously discovered [90].

Three genes (*folB*, *moaB* and *moaC*) are related to the metabolism of molybdenum cofactor. Genes containing this prosthetic group are formate dehydrogenase, nitrate reductase and some others [91]. Molybdopterin biosynthesis genes, including those in our metagenome (*moaB* and *moeB*), have been identified in the *Gilliamella* genome [92]. Members with these pathways have yet to be identified.

Petrobactin biosynthesis genes have been identified in presumably *Bacillus* reads (*yclN*, *yclB* and *yclP*). This siderophore is responsible for iron uptake in pathogenic bacteria [93]. In small hive beetles, the presence of *Bacillus licheniformis* was previously noted [59], some strains of which are capable of producing it.

Microcin biosynthesis genes *ycaO* and colicin biosynthesis genes *cvaA* are found in the metagenome. These antibiotics are produced by a variety of Bacilli.

Macrolide resistance gene *macB*, bicyclomycin *bcr* and bacitracin *bceA* have been found. *Streptomyces* from *Apis dorsata* bees and hive environments are known to produce antimicrobials [86]. Bacitracin is a well-known *Bacillus* antimicrobial. It is suggested to be found in honey [87].

All antimicrobial-related genes could be theoretically found in the “ABE” group species, first of all in Actinobacteria and *Bacillus*. It is these species that show the highest negative correlations with other groups, including the potentially pathogenic “S+S” and “*Pseudomonas*”. The “ABE” group is just as negatively correlated with “gut”, but much less so with “*Bartonella*”. This may be both an artifact of the analysis and an indication that “*Bartonella*” representatives have a separate function in this microbiome. Unlike other bacteria associated with the bee gut, members of this subgroup are facultative aerobes [60,82] and their development is probably less suppressed in combs.

*Apilactobacillus kunkeei* is a member of adult and larvae gut microbiota [13,80], but also was found in honey [35,78]. No significant correlations were found for it as well as for *Bifidobacterium*, *Clostridium* and some other species. These groups may include random microorganisms, for example—it should include closely related organisms and organisms with low abundance in samples, because they may appear and disappear randomly in the metagenome and its classifications. On the other hand, it will also include microorganisms with special functions that define some meaningful differences between different combs.

The composition of the microbiome varied little between individuals (Figure 2, Appendix A). All “ABE” group and fungi species represent 30–70% of the total classified signatures, and correlations between their amounts seems to be stable. Cluster analysis and PCA show differences between combs with larvae and pupae (Figure 4 and Figure 5).

Bee larvae guts contain many bacteria [15]. The composition of these communities is less constant than that of the adults [70], but it differs considerably from that found in their food [69,94]. Theoretically, the main sources of microbiota in larval combs should be communities from their gut or food. But it looks like brood combs carry an independent community, which may influence its own larvae communities. While some groups have similar amounts, the dominant group of larvae gut microbiome—Acetobacteraceae, *Parasacharibacter apium*—was detected in trace amounts.

How important is this microbiome in larval nutrition? A culture study predicts a significant shifting of the larval microbiome during adulthood [70]. No Actinobacteria were detected, and the overwhelming number of identified representatives belong to *Lactobacillus* and *Apilactobacillus*. On the other hand, gut examination by RT-PCR of prepupal larvae detects Actinobacteria in the gut of some larvae and pupae, but in much lower numbers than other bacteria [95]. DGGE analysis predicts much more consistency for larvae gut microbiome than pupal [71] and bigger amounts of proteobacteria (maybe *Gilliamella*) in pupal guts. In this study, the number of Actinobacteria and Gammaproteobacteria 16S in larval guts seems to be constant, while the Firmicutes amount grows to the sixth-instar larvae and then significantly decreases. The difference in *Apilactobacillus* amount and minor species composition between pupae and larvae combs in the present study must be an indicator of these processes.

It is known that the gut microbiota of pupae in bumblebees can influence changes in their phenotypes [96]. In bees themselves, an antagonistic effect of this microbiome on *Paenibacillus alvi* has been shown [97]. This may be due to the secondary metabolites it secretes and less likely to its composition because of its inconsistency. Microbes themselves increase antimicrobial response by the host bee [98].

The larval microbiome was previously thought to be populated by ingestion of contaminated food [69,94]. It appears that the comb microbiome may play a probiotic role in initiating the development of normal microflora. It is likely that the microbiome of larvae, and most likely from combs as well, does not directly affect the pupation process [98]. Major bacteria from the larval gut are not inhibited by royal jelly [70]. The influence of larval nutrition on the brood comb microbiome remains to be elucidated.

Any analysis except PCA does not reveal differences between combs with larvae and pupae (Figure 5). Perhaps this is an indicator of the permanence of the microbiome, which remains stable across most taxonomic units regardless of environmental factors. On the other hand, the complex differences detected by PCA may indicate some difference in their functioning. Major differences in the number of *Apilactobacillus* and *Gilliamella* unreliably distinguish combs with pupae and larvae (Figure 4), and these are the ones that define the first major components. Differences between brood combs are observed only from the second component onwards, indicating the role of the remaining diversity.

Before pupation, bees plug the comb [13,57]. Pupae do not feed and their gut is almost completely devoid of microbes, especially in the later stages [13,71]. Their comb community may be less rich due to fewer nutrients. On the other hand, larval combs contain fewer CFUs than hive surfaces [15,55]. Perhaps some species responsible for the constancy of their microbiome may disappear from the microflora, leading to an increase in diversity. Finally, a change in conditions may lead to the development of a new community altogether. Thus, the study of this community and its role in hive life is just beginning.

## 4. Materials and Methods

### 4.1. Sample Collection

Organisms for the description of surface microorganisms were collected from two different hives on 7 July 2022. Samples were collected from different frames. We used only 5th instar larvae and mostly white pupae. Every sampled comb was washed using Qiagen lysis buffer for DNA extraction (Qiagen, Hilden, Germany). Every comb was flushed twice, per 200 μL. For better cell destruction, samples were frozen and unfrozen for 3 times. Next day DNA from all samples were extracted using Blood & Cell Culture DNA Mini Kit (Qiagen, Hilden, Germany). All obtained samples were stored in individual Eppendorf Tubes at −20 °C.

### 4.2. Sequencing

Library preparation for shotgun metagenomics DNA was quantified and quality-assured with a capillary electrophoresis TapeStation 4200 (Agilent, Santa Clara, CA, USA), and sequenced using the Illumina HiSeq (San Diego, CA, USA). Expected read length was 100 bp. In total, we sequenced 17 comb materials from 2 different hives. Both hives are engaged in Russia, Leningrad oblast.

### 4.3. Data Processing

Reads were trimmed using Trimmomatic [99]. Then, contamination was removed by bowtie2 [100]. For decontamination of human DNA and removing host DNA we used the recent GRCh38 human genome and *Apis mellifera* genome GCA_003254395.2 from NCBI RefSeq.

### 4.4. Taxonomic Analysis

Taxonomic analysis was performed using Kraken 2 [101] and Kaiju [102] on RefSeq and HoloBee databases. All charts were plotted using a customized script using R programming language [103], RStudio [104] and necessary libraries [105,106,107,108,109,110,111,112,113]. Various metrics available in the R abdiv package [114] were used to analyze alpha diversity. Beta diversity was assessed using the Bray–Curtis and Jaccard dissimilarity coefficients.

IDBA-UD [115] assembly from pooled reads with *k*-mers 34–104, and a step size of 10 was used to analyze gene functions. Secondary metabolites and pathways were analyzed using Interproscan [116], antiSMASH [117], Prokka [118] with Artemis [119] and MinPath [120]. All results were summarized.

## 5. Conclusions

The brood combs microbiome exhibits distinct characteristics compared to other hive microbiomes, including honey. Most of the dominant species are aerobic saccharolytic bacteria. Although certain species may experience occasional outbreaks of diversity, the overall diversity remains remarkably stable and shows minimal variation between combs.

The differences observed between larval and pupal combs are primarily determined by the whole present community. It is possible that there are quantitative variations in metagenome abundance between these two types of combs, leading to the disappearance of some species with lower diversity from the annotations. This phenomenon could be attributed to the changing conditions between larval and pupal combs, such as the depletion of a constant source of new sugars and alterations in other environmental factors. Surprisingly, these changes do not significantly impact the underlying biodiversity of the combs. Therefore, it is likely that the minor diversity plays a crucial role in the functioning of this microbiome.

The contribution of the comb microbiome to the developmental trajectory of bees remains an enigma that requires further exploration. While the mechanisms involved in the metamorphosis of typical larvae into prospective hive queens are well understood and associated with nutritional factors, the precise mechanisms that regulate the decision-making process, determining which larvae will receive an adequate diet, remain obscure. It is possible that a selection protocol exists for larvae allocated to receive the royal jelly intended for emerging queens, which may be intertwined with the biochemistry of the comb microbiomes.

## Figures and Tables

**Figure 1 ijms-25-00741-f001:**
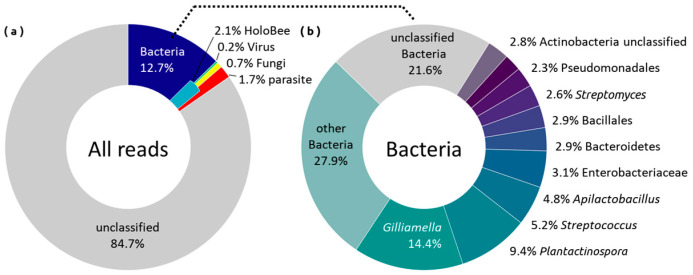
Results of taxonomic annotations with Kraken 2 on RefSeq database. (**a**): mean composition of all reads; (**b**): mean composition of bacterial reads.

**Figure 2 ijms-25-00741-f002:**
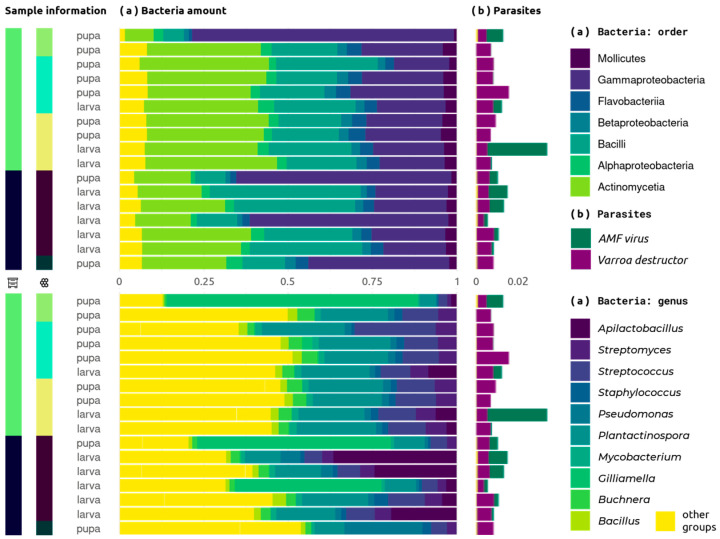
Composition of bacterial sequences in the microbiome based on Kraken 2 results. The side colors indicate the hive and sampling frame, while the column sizes correspond to the percentage of all reads. Top chart (**a**) shows the composition of different phylotypes, while the bottom (**b**) shows genus composition. Yellow corresponds to other species groups in all parts of the figure.

**Figure 3 ijms-25-00741-f003:**
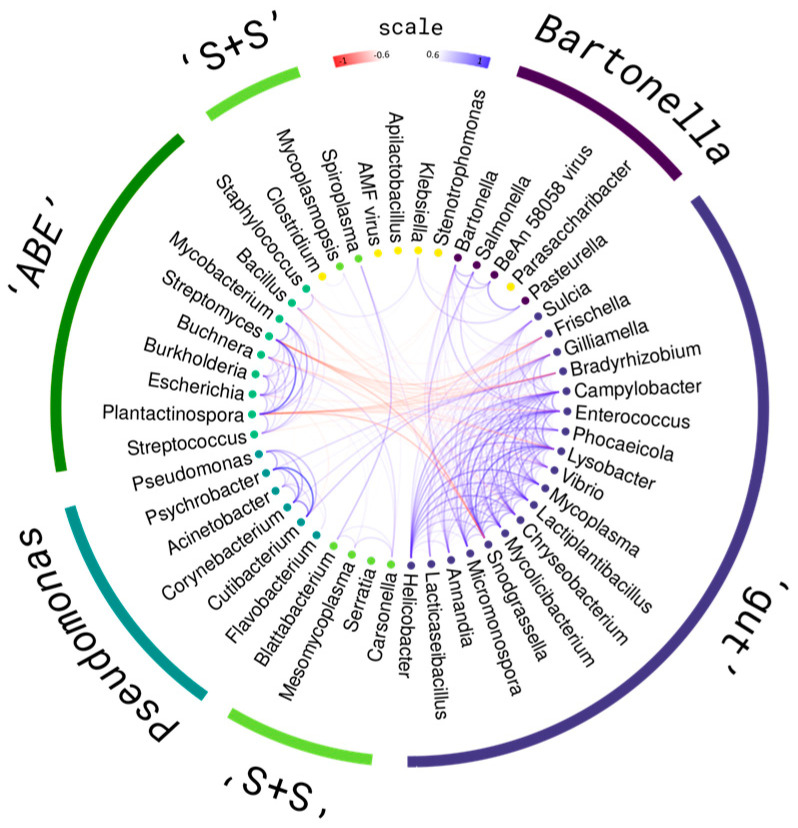
Edge bundling diagram illustrates the correlations between 50 most abundant genera. Color of each line represents Pearson correlation level between taxa abundances among classified samples. Circle colors describe clusters on correlation matrix using complete distance. On 7 *k*-mers, 5 groups show intergroup correlations with levels above 0.5 (Appendix A). Clusters named by the species with the highest amount. ‘Gut’ cluster is composed mostly of the genus previously described as a part of bee gut microbiota. ‘S+S’ is an abbreviation for *Spiroplasma* and *Serratia*. ‘ABE’ refers to the names of groups which include most genus of this cluster: Actinobacteria, Bacilliaceae and Enterococcacce. Members with weak correlations with other groups are indicated in yellow. Only correlations with absolute value bigger than 0.6 are shown.

**Figure 4 ijms-25-00741-f004:**
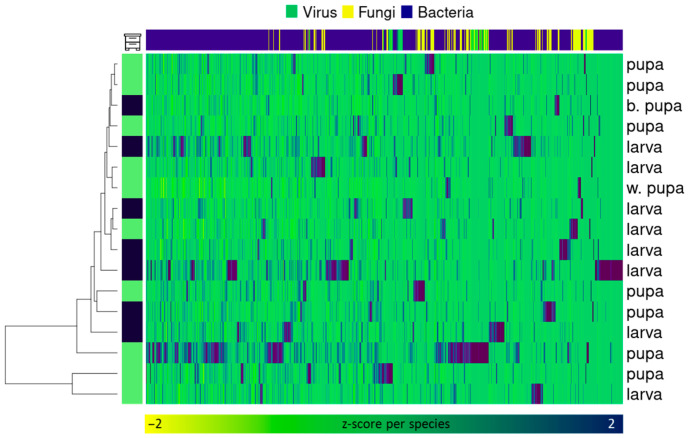
Heatmap on composition analysis. Dataset has been assembled from fungi genus, and bacteria and virus species amounts. The dendrogram was constructed using the ward.d2 method for samples and species both. Colors on the side indicate the hive and sampling frame. Colors on top indicate taxonomic affiliation. The map is colored according to z-score. b. pupa is an abbreviation for late = black pupa, and w. pupa is for early one.

**Figure 5 ijms-25-00741-f005:**
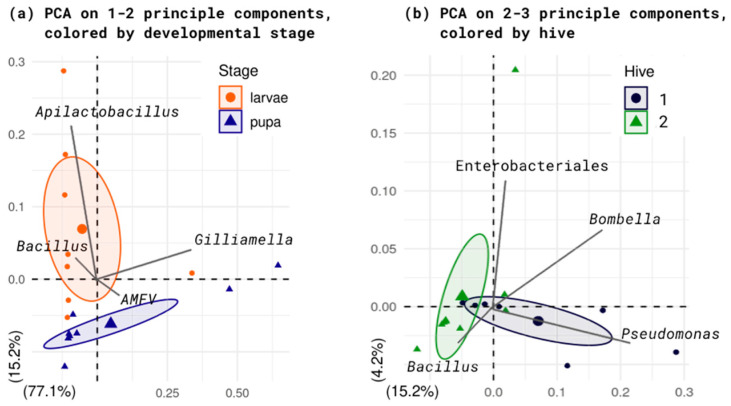
PCA of the metagenome compositions. (**a**): based on developmental stage; (**b**): based on hive. Dots and circles represent samples, while lines indicate vector projections of selected clades. Full dataset has been assembled from fungi and bacteria genus and virus species amounts.

## Data Availability

All data used in the manuscript was published at BioProject PRJNA1048732. For reproducibility and possible use in other studies, we have also added information about the code availability: R scripts for visualizations are available at https://github.com/dsmutin/aRchiteutis (accessed on 1 October 2023).

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
