# Peer review of "Shotgun Metagenomics Reveals Minor Micro“bee”omes Diversity Defining Differences between Larvae and Pupae Brood Combs"

_ijms, 2024, doi:10.3390/ijms25020741_

Round 1
Reviewer 1 Report
Comments and Suggestions for Authors
In this study, Daniil et al. investigated the differences in the microbiome composition of larvae and pupa brood combs using the shotgun sequencing method. Although the authors have done a lot of analysis of the metagenomic data, there are some areas that need further clarification. Here are some comments on this review:
1. It is proposed to reduce the number of references, 168 references were too many for a research paper.
2. Line 65 “on the other hand” appeared twice in one sentence.
3. The introduction needs to be revised and refined. The shortcomings of the current studies should be clearly presented, as well as the reasons why studying the microbiome compositions of larvae and pupa brood combs.
4. The genus names should be italicized, such as line 70 “Bacillus, Lactobacillus”.
5. Line 75 “10-106 CFU/g” should be 10-106.
6. I am a little confused about the definition of “diversity” in this study, especially in the discussion section. To my knowledge, diversity refers to community diversity, such as alpha diversity (Shannon index …) and beta diversity (bray-curtis …). Line 90 “2,1% of all diversity was classified on the HoloBee database”, could the authors provide any explanation of the meaning of the diversity?
7. Line 94 “OTUs” were OTUs clustered by 97% similarity? OTU is generally used for 16s rRNA gene sequencing analysis.
8. Lines 95-101, Given that the research topic of this paper is the microbiome composition of larvae and pupa brood combs, it is recommended that the evaluation of the metagenome analysis software be removed, in particular, “Tools often misclassify species [91]”.
9. It is recommended to create a flowchart of the sampling, it is better to let the readers know how the experiment was conducted.
10. Line 187 “Metagenome compositions are certainly important, but does it have functional sig- 187 nificance?” Suggest a different way of expressing it, not a doubtful sentence.
11. Line 193 “Results of bacterial diversity analysis” I do not feel it was diversity analysis.
12. Line 197 “The heatmap does not form any discernible patterns in species diversity.” Was it based on species or genus?
13. Figure 5 should be placed in the result section, not in the discussion section.
14. Line 381 “t -20*C”.
15. It is recommended that the authors rewrite the conclusion section to make it more refined.
Author Response
Dear reviewer,
We greatly appreciate the time and effort you have dedicated to reviewing our manuscript, and we would like to thank you for your constructive comments which have brought an important issue to our attention.
- It is proposed to reduce the number of references, 168 references were too many for a research paper.
Thank you for your suggestion to reduce the number of references in our manuscript. We recognize that the initial count of 168 references may appear excessive for a single research paper. While we agree that the Introduction section might exhibit an apparent surplus of citations, we wish to clarify that these references are not singularly used within this section but are repeatedly cited throughout the paper, particularly within the Discussion section.
Reducing the number of references cited in the Introduction would not alleviate the necessity to cite them later in the manuscript to substantiate our discussions and support our conclusions. Consequently, such a strategy would not significantly diminish the overall reference count. However, we have thoroughly analyzed the manuscript and successfully integrated certain quotations without compromising the scientific rigor and integrity of our work. It is important to note that a portion of the references to literary sources has been relocated to the supplementary materials file, as these references only indirectly pertained to the main body of the manuscript. This decision was made to maintain the focus and cohesiveness of the main text.
As a result of this revision, we have successfully decreased the number of references to a more focused ensemble of 148 sources. We trust this effort enhances the readability of the text while still providing comprehensive coverage of the relevant literature. We hope this amendment meets approval, and we are grateful for the opportunity to refine our manuscript accordingly.
- Line 65 “on the other hand” appeared twice in one sentence.
Thank you for noting the repetition of the phrase "on the other hand" in the same sentence on line 65. We appreciate your attention to detail. The revised sentence has been marked in yellow for ease of identification and to ensure clarity in the manuscript.
- The introduction needs to be revised and refined. The shortcomings of the current studies should be clearly presented, as well as the reasons why studying the microbiome compositions of larvae and pupa brood combs.
Thank you very much for your invaluable feedback on the introduction structure of our article. We have taken your comments into careful consideration and have made revisions to more clearly present the limitations of current research. We included information that, despite the high level of knowledge of the intestinal metagenome, the composition of the microbiome of larvae and pupal brood combs remains less studied, with studies primarily carried out using cultural methods. Additionally, we have revised the introduction to present a clear and brief summary of the current comprehension of the hive microbiota's functional elements. The revisions have been highlighted in yellow. We appreciate your valuable support in improving the influence of our research.
- The genus names should be italicized, such as line 70 “Bacillus, Lactobacillus”.
Thank you for your comment regarding the formatting of genus names in our manuscript. We concur with your observation that the genus names, such as "Bacillus" and "Lactobacillus" on line 70, should indeed be italicized in accordance with the standard conventions of scientific writing.
We regret that this typographical oversight was not caught during the final proofreading stage of our manuscript preparation. We understand the importance of adhering to proper scientific nomenclature, and we appreciate your attention to detail.
Please be assured that we will meticulously correct this error and ensure that all taxonomic names are presented in italics throughout the revised manuscript. All additional modifications in the text have been highlighted in yellow in the revised version of the manuscript.
- Line 75 “10-106 CFU/g” should be 10-106.
We extend our heartfelt appreciation for pointing out the typographical error in Line 75 of our manuscript. We have duly noted the error and have made the appropriate corrections in the revised version of the manuscript. We are grateful for your attention to detail.
- I am a little confused about the definition of “diversity” in this study, especially in the discussion section. To my knowledge, diversity refers to community diversity, such as alpha diversity (Shannon index …) and beta diversity (bray-curtis …). Line 90 “2,1% of all diversity was classified on the HoloBee database”, could the authors provide any explanation of the meaning of the diversity?
Thank you for this valuable comment. By the term diversity we mean, on the one hand, a list of species and, on the other hand, precisely the metrics that allow us to describe it. The point is that conventional methods of measuring alpha and beta diversity have rather low resolution. For the convenience of analyzing the article and to avoid terminological confusion, we tried to use this term more accurately. We have tried to divide the methods used into those that analyze diversity and those that analyze whole composition. We have also added a section devoted to diversity measures.
- Line 94 “OTUs” were OTUs clustered by 97% similarity? OTU is generally used for 16s rRNA gene sequencing analysis.
Thank you for your comment. This term sometimes is used in description of shotgun results classification, but maybe it is more relevant to use “genomic signatures” or taxa classification level here.
- Lines 95-101, Given that the research topic of this paper is the microbiome composition of larvae and pupa brood combs, it is recommended that the evaluation of the metagenome analysis software be removed, in particular, “Tools often misclassify species [91]”.
Thank you for the idea. We think that this section is really important for future works. Validation and experimentation with different classification approaches on our data were significantly hindered by data paucity on this theme. But for readers who are not going to use taxonomic classification tools on complicated shotgun metagenomes it could be an unnecessary part. So we made a decision to put it in supplementary as a comment - S3.
- It is recommended to create a flowchart of the sampling, it is better to let the readers know how the experiment was conducted.
We express our sincere gratitude for your suggestion to include a flowchart describing the sampling process. Such visual aids undeniably enhance the reader's understanding of the experimental design and methodology.
We would like to inform you that a flowchart detailing the sampling protocol has indeed been created and is intended to be presented as Figure 0 (Graphic Abstract) in our manuscript. We regret that, due to a possible oversight during the automatic typesetting of the draft manuscript, this crucial figure may not have been appropriately attached.
We will ensure that this matter is addressed with the editorial team and that the flowchart is duly included for clarity when we resubmit our manuscript.
- Line 187 “Metagenome compositions are certainly important, but does it have functional significance?” Suggest a different way of expressing it, not a doubtful sentence.
Thank you for your constructive feedback regarding the phrasing used in line 187 of our manuscript. Upon reflection, we recognize that the sentence in question may convey an unintended tone of uncertainty regarding the functional significance of metagenome compositions.
In accordance with your suggestion, we have rephrased the sentence to more affirmatively express the importance of functional implications stemming from metagenomic composition. The revised sentence now reads: "While metagenome compositions provide valuable insights, it is imperative to also consider their functional significance within the community".
We believe that this revision more directly addresses the critical nature of the functional aspects of metagenomic data and mirrors the intent of our discussion with greater clarity.
- Line 193 “Results of bacterial diversity analysis” I do not feel it was diversity analysis.
Thank you for this comment. It might be difficult to divide different analysis types, but for better understanding of our research we have changed it to “composition analysis” here and in some other places. For your convenience, the change has been highlighted in yellow.
- Line 197 “The heatmap does not form any discernible patterns in species diversity.” Was it based on species or genus? Даня
Thank you for finding this problem in the figure description. We add part about dataset processing to all related plots. For your convenience, the change has been highlighted in yellow.
- Figure 5 should be placed in the result section, not in the discussion section.
Thank you for your astute observation regarding the placement of Figure 5 in our manuscript. We appreciate your guidance in this matter.
In response to your comment, we have relocated Figure 5 to the results section. This change aligns the figure with the relevant data and findings, ensuring a more logical flow and coherence within the manuscript. We believe this adjustment enhances the clarity and readability of our paper.
- Line 381 “t -20*C”.
We agree with the suggested amendment and have incorporated the correction in the revised version of the manuscript. For your convenience, the change has been highlighted in yellow.
- It is recommended that the authors rewrite the conclusion section to make it more refined.
Thank you for your valuable feedback on our manuscript. Your constructive feedback has significantly improved the quality and clarity of our manuscript. We appreciate your suggestion to refine the conclusion section, and we agree that it required further attention to enhance its clarity and precision.
We have carefully reviewed and rewritten the conclusion section, incorporating your recommendations and believe that these revised conclusions provide a more refined summary of our study's key findings, highlighting the unique characteristics of the brood combs microbiome and setting the stage for future investigations into the intricate interactions between the microbiome and bee development.
We are eager to hear your thoughts on the revised manuscript and remain open to any further suggestions you might have that could refine our work even more.
Once again, thank you for your constructive feedback and for aiding us in elevating the quality of our research output.
Sincerely yours,
Adonin Leonid, Smutin Daniil and co-authors.

Reviewer 2 Report
Comments and Suggestions for Authors
Authors have studied brood comb microbiomes of honey bees. The article is confusing and lacks in several areas. Please use the following as constructive criticism.
Line 13-14: “model organism within contemporary microbiology” What makes bees such a model organism?
Add some quantitative data in abstract. How did the microbiomes of different stages differed? Was alpha diversity higher or lower at a particular life stage? Did they differ based on beta diversity?
Introduction is inadequate. It is not clear what the authors intend to study? Which research gaps is this article filling? If microbiome of bees is well-studied, then what is the relevance of this study? Why are certain bacteria italicized while others are not?
Materials and methods are inadequate for comparing differences between larval and pupal comb microbiomes. There is no mention of diversity metrics to evaluate these differences.
Results are purely descriptive and not analytical.
With all the above sections inadequate, it was hard for me to follow the discussion.
Author Response
Dear reviewer,
First and foremost, We would like to express my sincere gratitude for your insightful comments and the time you have devoted to reviewing our manuscript. Your expertise and meticulous attention have been invaluable in guiding us toward enhancing our work.
- Line 13-14: “model organism within contemporary microbiology” What makes bees such a model organism?
In response to the comment on lines 13-14 regarding the use of bees as a "model organism within contemporary microbiology," it is important to clarify the attributes that render Apis mellifera, or the honeybee, an exemplary model organism for certain microbiological studies. The honeybee presents a unique symbiotic relationship with its microbiota, harboring a specialized and relatively simple gut microbial community that is distinct from those found in other insects and vertebrates. This simplification allows for the dissection of microbial interactions and their influence on the host in a more controlled and interpretable manner compared to more complex microbiomes.
Furthermore, bees are of significant ecological and agricultural importance due to their role in pollination, making them a subject of extensive study in terms of their health, behavior, and survival. The decline in bee populations has spurred concerted research efforts aiming to understand the underlying causes, among which the role of microbes and pathogens is paramount.
From a practical perspective, honeybees are also relatively easy to rear and manipulate in experimental settings, and their colonies can be maintained in both field and laboratory conditions. This provides researchers with the flexibility to conduct a range of experiments, from controlled laboratory assays to field-based ecological studies.
Finally, the sequencing of the honeybee genome and the availability of molecular tools have enabled insights into the genetic basis of bee-microbe interactions and the impact on bee physiology and immunity. Collectively, these aspects contribute to the status of honeybees as a model organism in microbiological studies, particularly in the context of host-microbe interactions, ecosystem function, and health research.
- Add some quantitative data in abstract. How did the microbiomes of different stages differed? Was alpha diversity higher or lower at a particular life stage? Did they differ based on beta diversity?
We appreciate your suggestion. It is hard to find a suitable quantitative metric for diversity based on the shotgun metagenome, but we try to apply it and add some necessary facts to our abstract, and a section in results dedicated to diversity quantitative measures. We hope that these changes help to make easier and faster insights on our article.
- Introduction is inadequate. It is not clear what the authors intend to study? Which research gaps is this article filling? If microbiome of bees is well-studied, then what is the relevance of this study? Why are certain bacteria italicized while others are not? Даня
Thank you for this comment. To provide a better understanding of the modern situation on our topic, we have reduced and split the introduction in two parts.
Bee microbiome in whole is well-studied, and it is one of the reasons to study remaining parts. Comb microbiota was previously described only based on cultural methods. It is indispensable to use novel approaches to reveal new insights on work of bee hives. Members of this microbiome play a significant role as a part of bee superorganism. Some of them were not previously described as hive communities members and can influence bee colony health, production efficiency and development processes.
We concur with your observation that the genus names should indeed be italicized in accordance with the standard conventions of scientific writing. We regret that this typographical oversight was not caught during the final proofreading stage of our manuscript preparation. We understand the importance of adhering to proper scientific nomenclature, and we appreciate your attention to detail.
Please be assured that we will meticulously correct this error and ensure that all taxonomic names are presented in italics throughout the revised manuscript. All additional modifications in the text have been highlighted in yellow in the revised version of the manuscript.
- Materials and methods are inadequate for comparing differences between larval and pupal comb microbiomes. There is no mention of diversity metrics to evaluate these differences.
We appreciate valuable feedback regarding the adequacy of our materials and methods section for comparing differences between larval and pupal comb microbiomes. We acknowledge that the initial version of our manuscript did not adequately address the use of diversity metrics in evaluating these differences.
In response to this concern, we have thoroughly revised the structure of the methods section and included additional information on the bioinformatics tools employed to assess and compare microbial diversity. Specifically, we have incorporated details on the specific diversity metrics utilized, such as alpha and beta diversity indices, to accurately quantify and analyze the variations between larval and pupal comb microbiomes.
By including this data, we have guaranteed a thorough and sound method for assessing the distinctions in microbial makeup between larval and pupal comb samples.
- Results are purely descriptive and not analytical.
Thank you for your thoughtful feedback concerning the presentation of results in our manuscript. We wish to clarify that the structure of our manuscript adheres to the classical scheme, which distinctly segregates the Results and Discussion sections. While we appreciate that some journals favor a combined Results and Discussion section, allowing for immediate analytical commentary, our chosen architecture maintains a clear separation of these sections.
In keeping with this traditional format, we have delineated the Results section to provide a straightforward account of our empirical findings. Subsequently, the Discussion section is reserved for a thorough analysis of these results, where we contextualize our findings within the broader corpus of existing scientific literature.
We concur with your observation that integrating analysis within the Results might be conducive in certain publishing contexts, and we are grateful for the opportunity to explain the rationale behind our chosen format. We assure you that a comprehensive analytical examination of our results is indeed encompassed in the Discussion section, aligning with the structural flow of our article.
- With all the above sections inadequate, it was hard for me to follow the discussion.
Regarding your observation that the inadequacies in the various sections made it challenging to follow the discussion, we concur with your assessment and understand the importance of a coherent narrative for the smooth comprehension of our research findings. We have taken your critique to heart and, combined with advice from another esteemed reviewer, have meticulously revised the structure and logic of our manuscript to ensure a more seamless and intuitive reading experience.
It is with hopeful anticipation that we have managed to address the concerns you have raised, thereby facilitating a smoother and more engaging interaction with the material presented. We believe that the amendments we have implemented will resonate with your academic standards, and we exceedingly hope the manuscript now meets your approval and the esteemed IJMS's rigorous requirements.
We are eager to hear your thoughts on the revised manuscript and remain open to any further suggestions you might have that could refine our work even more.
Once again, thank you for your constructive feedback and for aiding us in elevating the quality of our research output.
Sincerely yours,
Adonin Leonid, Smutin Daniil and co-authors.

Round 2
Reviewer 1 Report
Comments and Suggestions for Authors
Thank you for the authors's reply!
I have no further concerns.
Author Response
Dear reviewer,
We greatly appreciate the time and effort you have dedicated to reviewing our manuscript, and we would like to thank you for your constructive comments which have brought an important issue to our attention.
All edits from first review round is highlighted yellow; all novel changes are marked blue.
- Numbering of the supplementary material is confusing. S3 exists as a figure and as a comment. Please use Figure S1, Fig. S2 etc. in the manuscripts as well as in the Supplementary Material. For the comment say Comment S1 in both cases.
Thank you for the clarification on the format of the supplementary mention. They have been taken into account and all necessary corrections have been made.
- gene names must be written in italics in the whole manuscript
Thanks for noticing that typo, it appeared when transferring between documents. It is fixed.
- line 98: 2.1%
Thanks for noticing that format error. It is fixed.
- Fig. 3: explain all acronyms used in the figure in the figure legend (not only in the text)
Thank you for this comment. We thought about it when inserting the picture into the article. The description of the clusters is rather cumbersome and possibly overloads the legend. We tried to explain the proposed names as briefly as possible.
- line 194 and others: give all species names in italics
Thanks for noticing that typo, it appeared when transferring between documents. We have reread our text attentively and applied all necessary format changes.
- line 351: fungi in lowercase letters line 353 and others: the plural of larva is larvae and of pupa is pupae
Thanks for noticing that grammar error. It is fixed in both article text and the title.
- line 354: either say gut contains or guts contain
Thanks for noticing that grammar error. It is fixed.
- line 365 and others: why here Actonibacteria etc. in lowercase letters?
Thanks for noticing that format error. It is fixed.
- line 419: analysis was performed
Thanks for noticing that grammar error. It is fixed.
- line 449: give here the supplementary material shown in your paper
We apologize for the oversight. We will promptly provide the information of supplementary material for line 449 as requested.
- References must be carefully checked: species names in italics, all paper title in lowercase letters etc.
Thank you for that mark. As there was a lot of work to be done, we used an automatically compiled reference list and there may have been formatting errors in some places. Corrected.
- line 729: Staphylococcus
Thanks for this much attention, even to . Edited.
- the number of references (148 in the text and 17 in the Supplementary Material) is still too high for an original paper.
In the current edition we have tried to minimize literature citations as much as possible, primarily in the introduction. It now contains 58 sources, a significant part of which is also used in the results and their discussion (61 sources). 22 references refer to the bioinformatic programs used. Further reduction, in our opinion, may harm the validity of the narrative.
We are eager to hear your thoughts on the revised manuscript and remain open to any further suggestions you might have that could refine our work even more.
Once again, thank you for your constructive feedback and for aiding us in elevating the quality of our research output.
Sincerely yours,
Adonin Leonid, Smutin Daniil and co-authors.
